# Clinical Outcomes for Primary and Radiation-Associated Angiosarcoma of the Breast with Multimodal Treatment: Long-Term Survival Is Achievable

**DOI:** 10.3390/cancers13153814

**Published:** 2021-07-29

**Authors:** Joshua P. Kronenfeld, Jessica S. Crystal, Emily L. Ryon, Sina Yadegarynia, Celeste Chitters, Raphael Yechieli, Gina D’Amato, Andrew E. Rosenberg, Susan B. Kesmodel, Jonathan C. Trent, Neha Goel

**Affiliations:** 1Sylvester Comprehensive Cancer Center, Division of Surgical Oncology, Department of Surgery, University of Miami Miller School of Medicine, Miami, FL 33132, USA; joshua.kronenfeld@jhsmiami.org (J.P.K.); jessica.crystal@med.miami.edu (J.S.C.); emily.ryon@jhsmiami.org (E.L.R.); sina@med.miami.edu (S.Y.); cec240@med.miami.edu (C.C.); sxk1006@med.miami.edu (S.B.K.); 2Sylvester Comprehensive Cancer Center, Department of Radiation Oncology, University of Miami Miller School of Medicine, Miami, FL 33132, USA; ryechieli@med.miami.edu; 3Sylvester Comprehensive Cancer Center, Division of Medical Oncology, Department of Medicine, University of Miami Miller School of Medicine, Miami, FL 33132, USA; gina.damato@med.miami.edu (G.D.); jtrent@med.miami.edu (J.C.T.); 4Sylvester Comprehensive Cancer Center, Division of Anatomic Pathology Services, Department of Pathology & Laboratory Medicine, University of Miami Miller School of Medicine, Miami, FL 33132, USA; arosenberg@med.miami.edu

**Keywords:** breast angiosarcoma, multimodality therapy, neoadjuvant chemotherapy, pathologic complete response

## Abstract

**Simple Summary:**

Primary angiosarcoma of the breast (PAS) and radiation-associated angiosarcoma of the breast (RAAS) are rare sarcomas that affect the inner lining of blood vessels in the breast with incidences of 0.05% and 0.02%, respectively. This study demonstrates that multimodal treatment with neoadjuvant chemotherapy, radiation therapy (in some patients), surgery, and adjuvant chemotherapy may result in optimal clinical outcomes, including prolonged survival. While this study reflects a small sample size, it demonstrates that neoadjuvant chemotherapy, especially when associated with a pathologic complete response, may contribute to these substantial results. Breast angiosarcoma, however, remains a rare disease and prospective, multi-institutional studies need to be performed to overcome the inherent limitations associated with low incidence.

**Abstract:**

Background: The optimal management of primary angiosarcoma (PAS) and radiation-associated angiosarcoma (RAAS) of the breast remains undefined. Available data show persistently poor survival outcomes following treatment with surgery or chemotherapy alone. The objective of this study was to evaluate long-term outcomes in patients treated with multimodality therapy. Methods: Patients diagnosed with stage I–III PAS or RAAS of the breast were identified from our local tumor registry (2010–2020). Patient demographics, tumor characteristics, and treatment were collected. Primary outcomes were local recurrence (LR), distant recurrence (DR), and median overall survival (OS). A secondary outcome was pathologic complete response (pCR) following neoadjuvant chemotherapy (NAC). Mann–Whitney U, chi-squared, or Fisher exact tests were used to analyze data. Kaplan–Meier curves compared OS for PAS and RAAS. Results: Twenty-two patients met inclusion criteria, including 11 (50%) with RAAS and 11 (50%) with PAS. Compared to PAS patients, RAAS patients were older and had more comorbidities. For RAAS patients, median time from radiation to diagnosis was 6 years (IQR: 5–11). RAAS patients were more likely to have a pCR to NAC (40% vs. 20%, *p* = 0.72). RAAS patients had a higher LR rate (43% vs. 38%, *p* = 0.83), and PAS patients were more likely to develop a DR (38% vs. 0%, *p* = 0.07). Median OS was 81 months in PAS patients and 90 months in RAAS patients (*p* = 1.00). Discussion: Long-term survival can be achieved in patients with PAS and RAAS who undergo multimodality treatment. NAC can result in pCR. The long-term clinical implications of pCR warrant further investigation.

## 1. Introduction

Primary angiosarcoma of the breast (PAS) and radiation-associated angiosarcoma of the breast (RAAS) are rare sarcomas that affect the inner lining of blood vessels in the breast with incidences of 0.05% and 0.02%, respectively [1,2,3]. Moreover, given the aggressive nature of these sarcomas, locoregional recurrence rates (53% for PAS and 46% for RAAS) and distant metastasis (34% for PAS and 13% for RAAS) remain high, leading to poor median overall survival (OS) rates at 16 months for PAS and 12 months for RAAS following either surgery or chemotherapy alone [2,4,5,6,7,8]. As a result of low incidence and poor outcomes, optimal treatment guidelines remain limited.

Similar to the management of other cancers, recent studies have indicated that multimodality treatment including surgery, systemic chemotherapy, and radiation may be important in improving PAS and RAAS outcomes [6,9,10,11]. This paradigm shift from a surgery-only approach to a multidisciplinary approach has resulted in improved local control and decreased risk of distant metastasis at 68% and 38% for PAS and RAAS, respectively [3,12,13,14]. More recently, case reports and case series have evaluated the role of neoadjuvant chemotherapy with promising results [7,15].

Our National Cancer Institute (NCI)-designated Cancer Center (Sylvester Comprehensive Cancer Center) is a high-volume sarcoma referral center with a weekly multidisciplinary sarcoma tumor board. As a result, the primary objective of this study was to analyze local recurrence (LR), distant recurrence (DR), and survival outcomes in the setting of contemporary multimodality therapy for PAS and RAAS. The secondary objective was to evaluate pathologic complete response (pCR) rates after neoadjuvant chemotherapy (NAC) given our institutional preference to treat PAS and RAAS with NAC.

## 2. Materials and Methods

### 2.1. Data Source and Patient Selection

Female patients with diagnoses of PAS or RAAS between 2010–2020 were identified through our institutional tumor registry using International Classification of Diseases 9th edition codes (171.4 and 171.9). PAS or RAAS was confirmed based on institutional pathology review by a sarcoma pathologist. A total of 22 patients met inclusion criteria and were included in the study (Figure 1). 

Patients who did not receive all or part of their first course of treatment at our institution were excluded since complete clinical, treatment, and outcome data were not available in our electronic medical records. The study was approved by our institutional review board and need for informed consent was waived due to the retrospective nature of the study.

### 2.2. Variables

Patient, tumor, and treatment characteristics were retrospectively collected from electronic medical records. Data variables included age at diagnosis, sociodemographics (e.g., race, ethnicity, and health insurance), medical history (e.g., body mass index, Charlson Comorbidity Index (CCI—a risk score predictive of mortality based on specified comorbidities), and smoking status), prior breast cancer history with adjuvant radiation (e.g., yes or no), tumor characteristics (e.g., clinical stage, time from radiation exposure to diagnosis), treatment (e.g., receipt of NAC and adjuvant chemotherapy (regimen and cycles), receipt of neoadjuvant and adjuvant radiation, definitive surgical procedure (e.g., chest wall resection, modified radial mastectomy, total mastectomy, or partial mastectomy), and outcome data (e.g., local recurrence, distant recurrence, and OS). While some variables only had a few patients per category (e.g., one Asian patient, one Medicaid patient, one patient with total mastectomy as initial operation, or one chest wall resection at definitive surgery), the authors chose to not exclude any patients in order to detail the entire population treated at this center during the study period.

### 2.3. Statistical Analysis

Multiple group comparisons were performed for the above variables between PAS and RAAS cohorts. Fisher exact tests were used to compare categorical variables, and Mann–Whitney U non-parametric tests were used to compare continuous variables. These tests were utilized due to the relatively small sample size. Overall survival (OS) and recurrence-free survival (RFS, time from completion of primary treatment to recurrence) were analyzed for patients whose diagnoses were made prior to 2016 to allow for a minimum of a 5-year follow-up. Kaplan–Meier curves were generated and analyzed using log-rank tests. A 5-year follow-up was selected to evaluate if patients undergoing multimodality treatment could achieve long-term survival for PAS and RAAS, a sign of clinical progress in a disease where historical survival has been found to be less than 18 months [2,4,5,6,7,8]. A subgroup analysis was performed for all patients who underwent NAC to examine sociodemographics, NAC regimens, and pathologic outcomes. All *p* values were from two-sided tests and results were deemed statistically significant at *p* < 0.05. Statistical analysis was performed using SPSS version 25 (IBM Corporation, Armonk, NY, USA, copyright 2017) [16].

## 3. Results

### 3.1. Patient, Tumor, and Treatment Characteristics

A total of 22 patients were identified. Eleven (50%) had RAAS, and eleven (50%) had PAS (Figure 1). All patients were female with a median age of 70 years for RAAS patients and 52 years for PAS patients. RAAS patients were more likely to have comorbidities with 63.6% of all patients having a CCI score > 2 (Table 1).

Ten (91%) out of the eleven patients who developed RAAS initially had a partial mastectomy for their breast cancer with adjuvant radiation, while one patient had a mastectomy with adjuvant radiation. Patients with RAAS were diagnosed at a median of 6 years (interquartile range (IQR): 5–11) after completion of adjuvant radiation therapy. There was no difference in clinical tumor size between PAS and RAAS based on computed tomography or magnetic resonance imaging (4.2 vs. 2.8 cm, *p* = 1.00) (Table 1).

There was no significant difference in receipt of NAC or type of NAC regimens. A greater proportion of PAS patients received neoadjuvant radiation therapy compared to the RAAS cohort (18 vs. 0%, *p* = 0.14). Definitive surgical treatment included total or partial mastectomy, modified radical mastectomy, and chest wall resection. The predominant operation for PAS and RAAS patients was a total mastectomy (46 and 73%) and a greater percentage of PAS patients had a partial mastectomy (36 vs. 0%). One RAAS patient required a chest wall resection. There was no statistically significant difference in the median pathologic tumor size or an R0 resection margin. With regard to the final pathology, RAAS tumors were more likely than PAS to be high-grade (grades II–III) tumors (82 vs. 55%, *p* = 0.17). There was no difference between PAS and RAAS cohorts in receipt of adjuvant chemotherapy (46 vs. 46%, *p* = 1.00), type of adjuvant chemotherapy regimen (*p* = 1.00), or receipt of adjuvant radiation therapy (27 vs. 9%, *p* = 0.27) (Table 1).

### 3.2. Recurrence and Survival Information

Fifteen patients diagnosed with PAS or RAAS before 2016 with at least 5 years of follow-up were analyzed for LR, DR, RFS, and OS (Figure 1). RAAS patients were more likely to have an LR (43 vs. 38%, *p* = 0.83), while PAS patients were more likely to have a DR (38 vs. 0%, *p* = 0.07, Table 1). There was no difference between PAS and RAAS cohorts in OS (81 vs. 90 months, *p* = 1.00) (Table 1 and Figure 2) or RFS (55 vs. 53 months, *p* = 0.32) (Table 1 and Figure 3).

### 3.3. Subgroup Analysis of Neoadjuvant Chemotherapy Patients

Subgroup analysis for NAC patients (*n* = 10 patients, PAS (5, 50%) and RAAS (5, 50%)) was performed (Table 2). The patients received either double-agent therapy, consisting of doxorubicin/ifosfamide or gemcitabine/docetaxel, or single-agent paclitaxel. Patients received a median of four cycles of chemotherapy. A total of eight patients (80%) had a response. Of those, five patients (50%) had a partial response and three patients (30%) had a pCR. Response rates were not associated with type of angiosarcoma or other patient, pathologic, or treatment factors.

## 4. Discussion

This study shows that in the setting of contemporary multimodality treatment, patients diagnosed with and treated for breast PAS or RAAS can achieve prolonged recurrence-free and long-term survival. A paradigm shift from a surgical-only approach to the use of multiple different types of therapies, including NAC, may have contributed to the improved survival outcomes in our population. Specifically, PAS patients had a 5-year survival of 50%, while the RAAS patients had a 5-year survival of 43%. This is similar to a recent Surveillance Epidemiology and End Results (SEER) study looking at 313 patients with PAS and 591 RAAS patients, which showed 5- and 10-year survival rates of 49% and 38%, respectively, in PAS patients and of 42% and 27%, respectively, in RAAS patients [17].

Given that angiosarcoma of the breast is a rare diagnosis, there are limited data supporting a definitive treatment plan. As such, treatment recommendations for PAS and RAAS are based upon case series and institutional experiences [7,18,19]. Historically, surgical resection has been the mainstay of treatment for breast angiosarcoma, with partial mastectomy considered for smaller tumors and a total mastectomy for larger tumors [20]. In this study, most patients were treated with a partial or total mastectomy, but few patients required more radical resections, including a modified radical mastectomy or chest wall resection. While surgery remained a key component of therapy, all patients in our study ultimately received multimodality therapy.

Specifically, 50% received NAC, which has become increasingly utilized in the multimodal treatment of PAS and RAAS [14]. While studies are limited to small-sample investigations and case reports, a growing body of evidence supports NAC as reflected by the increased pCR rates [21,22]. Specifically, 80% of patients who received NAC achieved a partial response and 30% had a pCR. A study by Oxenberg et al. also reported a similar pCR rate in PAS patients; however, their study included cutaneous angiosarcoma in other locations as well as the breast [23]. While their study did not show a survival benefit, it did show that NAC was well-tolerated and associated with improved primary wound closure. More recent studies have demonstrated potential survival benefits in patients who achieve a pCR from NAC treatment of soft-tissue sarcomas [24]. While this previous study was also not exclusively examining breast angiosarcoma patients, it demonstrates that favorable response rates after NAC may portend improved OS. Attempts to further enhance response in the neoadjuvant setting have been made with the addition of hyper-fractionated radiation with hyperthermia [25]. Neoadjuvant radiation therapy has also been considered in PAS patients to decrease local recurrence, although the survival benefits remain unclear [7,26].

The addition of adjuvant chemotherapy and radiation has also been shown to benefit patients. Paclitaxel-based chemotherapies have been shown to delay progression of metastatic lesions and may prevent local recurrence in some patients [27,28]. Our study investigating the treatment and outcomes of PAS and RAAS patients over a decade is a robust experience that may help to guide future treatments and clinical trials. Adjuvant chemotherapy was delivered in nearly half of the patients in this study while only about 20% received adjuvant radiation. The patients included in this study received this therapy as part of a multimodality treatment plan developed following discussion at the regularly scheduled multidisciplinary sarcoma tumor board at a high-volume sarcoma cancer center. This treatment strategy may have also contributed to the long-term OS that was achieved in this cohort. Interestingly, PAS and RAAS patients had similar outcomes in this study and this may have been due to the standardized treatment regimen followed by treatment teams at this institution. Some studies have alluded to worse outcomes for RAAS patients; however, this was not found in this cohort [29].

As patients continue to be treated for complex malignancies such as PAS and RAAS, additional data will help to uncover the most effective multimodality treatment regimen. Multicenter prospective randomized control trials are needed to determine the utility of these therapies, and the patients described in this article will need to be followed for several more years to determine the long-term efficacy of our treatments. A final discussion point with regard to future directions is the potential benefits of sequencing the tumor by next-generation whole exome or RNA sequencing. Through RNA sequencing, the Angiosarcoma Project has noted mutations unique to tumors at different anatomic locations [30]. For example, in breast angiosarcoma, PIK3CA-activating mutations are observed in many tumors, which can likely serve as therapeutic targets [30]. Continued work to identify mutations specific to breast angiosarcoma can help to drive novel treatments that may facilitate improved patient outcomes [31]. Additionally, our group has identified actionable mutations, evaluated tumor mutational burden, and is assessing the efficacy of checkpoint inhibition [32,33]. This work lays the foundation for targeted therapy to tailor treatments for PAS and RAAS patients in the curative setting. Ultimately, we hope that modifying treatment regimens to reflect tumor biology will translate into more favorable results and improved long-term survival in these patients [34,35,36].

Overall, our study expands current data and highlights the benefits of using multimodality treatment to achieve long-term survival and of using NAC to achieve pCR. However, it is limited by its retrospective nature; specifically, unmeasured factors that may influence treatment decision-making. We were further limited in the small number of cases and likely underpowered statistical analyses, limiting our ability to effectively perform subgroup analyses (e.g., the impact of menopause on patient outcomes). Additionally, this study is limited in its generalizability as it was conducted at an NCI-designated cancer center with a high volume of sarcoma patients treated in a multidisciplinary setting. While the conclusions garnered from this study may be helpful to physicians at other centers, the results may be difficult to replicate without a multidisciplinary sarcoma team. Breast angiosarcoma, however, is a rare diagnosis and prospective multi-institutional studies need to be performed to overcome the inherent limitations associated with the low incidence of this diagnosis.

## 5. Conclusions

The use of multimodality treatment with chemotherapy, radiation treatment, and surgical interventions may result in long-term survival in patients with PAS and RAAS. Moreover, delivery of NAC may play a pivotal role in these favorable outcomes. Future long-term studies assessing the durability of pCR and even partial response will need to be performed to learn the optimal approach to these patients.

## Figures and Tables

**Figure 1 cancers-13-03814-f001:**
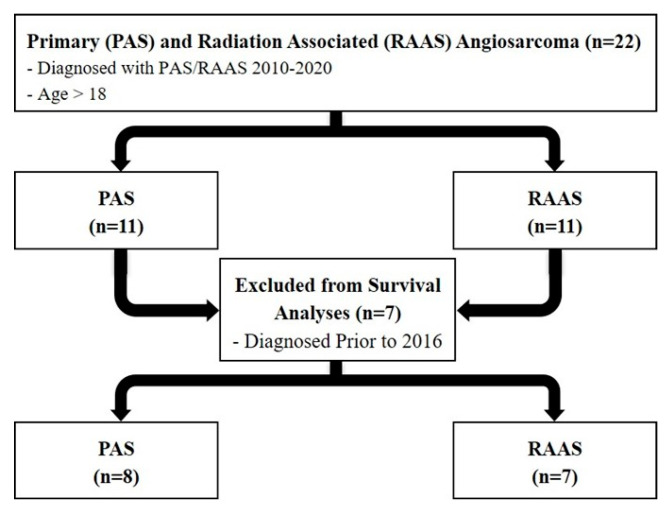
Study Design.

**Figure 2 cancers-13-03814-f002:**
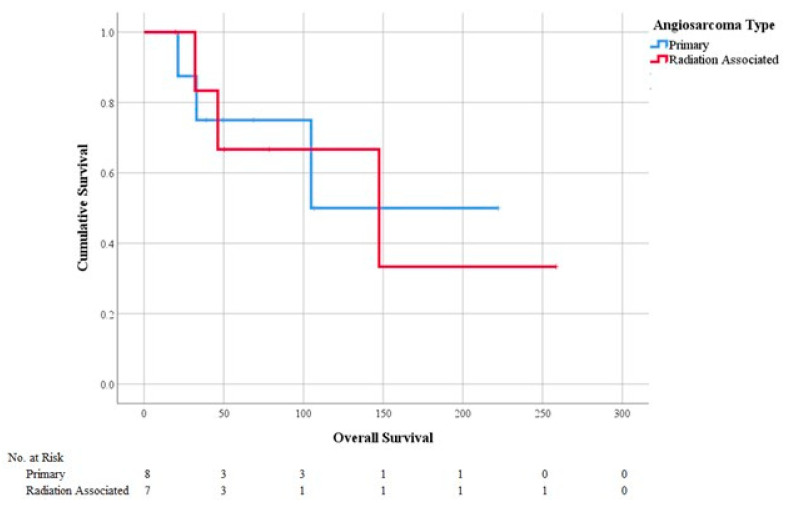
Overall survival by angiosarcoma type.

**Figure 3 cancers-13-03814-f003:**
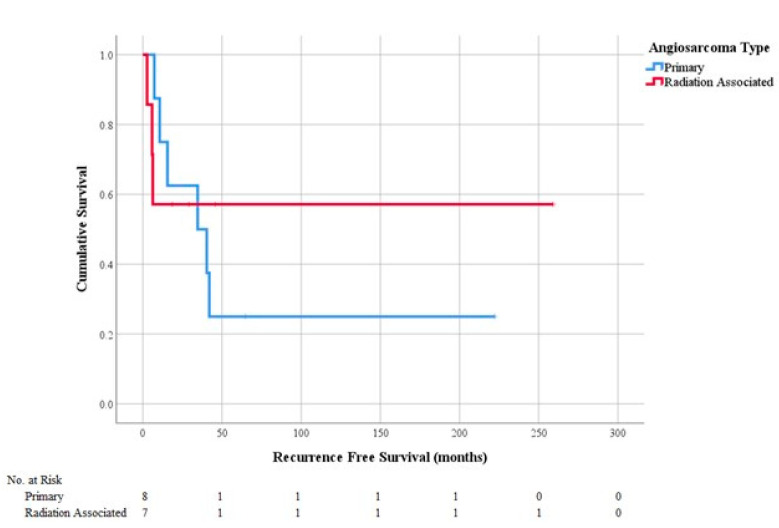
Recurrence free survival by angiosarcoma type.

**Table 1 cancers-13-03814-t001:** Patient, tumor, and treatment characteristics.

Variable		Breast Angiosarcoma Type			
		**Combined**	**Primary** **(PAS)**	**Radiation-Associated (RAAS)**	
		**(*n* = 22)**	**(*n* = 11)**	**(*n* = 11)**	
		***n*** **(%)**	***n*** **(%)**	***n*** **(%)**	***p***
Sociodemographics					
Age (median, IQR)		59 (50–72)	52 (43–68)	70 (57–76)	0.086
Race	*Asian*	1 (4.5)	0 (0.0)	1 (9.1)	0.306
	*Black*	0 (0.0)	0 (0.0)	0 (0.0)	
	*White*	21 (95.5)	11 (100.0)	10 (90.9)	
Ethnicity	*Hispanic*	9 (40.9)	7 (63.6)	2 (18.2)	**0.030**
	*Not Hispanic*	13 (59.1)	4 (36.4)	9 (81.8)	
Health Insurance	*Medicaid*	1 (4.5)	1 (9.1)	0 (0.0)	0.392
	*Medicare*	7 (31.8)	2 (18.2)	5 (45.5)	
	*Private*	11 (50.0)	7 (63.6)	4 (36.4)	
	*Uninsured*	3 (13.6)	1 (9.1)	2 (18.2)	
Health Information					
BMI (median, IQR)		24.5 (20.6–29.1)	24.6 (20.3–36.4)	24.3 (20.6–27.8)	--
Charlson Comorbidity Index	*0*	4 (18.2)	4 (36.4)	0 (0.0)	**0.023**
	*1–2*	4 (18.2)	3 (27.3)	1 (9.1)	
	*>2*	14 (63.6)	4 (36.4)	10 (90.9)	
Diabetes Mellitus		3 (11.6)	1 (9.1)	2 (18.2)	0.534
Blood Glucose Level at Diagnosis (median, IQR)		94 (88–104)	90 (89–95)	98 (86–108)	0.370
Menopause		11 (50.0)	3 (27.3)	8 (72.7)	**0.033**
Original Breast Cancer Information					
Initial Breast Surgery	*Partial Mastectomy*	10 (45.5)	0 (0.0)	10 (90.9)	**<0.001**
	*Total Mastectomy*	1 (4.5)	0 (0.0)	1 (9.1)	
	*No Operation*	11 (50.0)	11 (100.0)	0 (0.0)	
Angiosarcoma Diagnosis					
Radiation Exposure to Diagnosis in Years (median, IQR)		6 (5–11)	--	6 (5–11)	--
Clinical Tumor Size in cm (median, IQR)		3.5 (2.2–7.0)	4.2 (2.5–7.2)	2.8 (2.0–7.0)	--
Angiosarcoma Neoadjuvant Therapy					
Neoadjuvant Chemotherapy		10 (45.5)	5 (45.5)	5 (45.5)	--
Neoadjuvant Chemotherapy Regimen	*Doxorubicin/Ifosfamide*	5 (22.7)	3 (60.0)	2 (40.0)	0.469
	*Gemcitabine/Docetaxel*	3 (13.6)	2 (40.0)	1 (20.0)	
	*Single-Agent Paclitaxel*	2 (9.1)	0 (0.0)	2 (40.0)	
Neoadjuvant Chemotherapy Cycles (median, IQR)		4 (3–5)	5 (3–6)	4 (4–4)	0.167
Neoadjuvant Radiation		2 (9.1)	2 (18.2)	0 (0.0)	0.138
Angiosarcoma Definitive Surgery and Pathology					
Definitive Surgery (for PAS or RAAS)	*Chest Wall Resection*	1 (4.5)	0 (0.0)	1 (9.1)	0.128
	*Modified Radical Mastectomy*	4 (18.2)	2 (18.2)	2 (18.2)	
	*Partial Mastectomy*	4 (18.2)	4 (36.4)	0 (0.0)	
	*Total Mastectomy*	13 (59.1)	5 (45.5)	8 (72.7)	
Pathologic Tumor Size in cm (median, IQR)		4.3 (2.5–7.6)	4.5 (2.5–7.6)	4.0 (2.0–8.0)	0.395
Microscopic Margin Status	*R0*	20 (90.9)	9 (81.8)	11 (100.0)	0.138
	*R1*	2 (9.1)	2 (18.2)	0 (0.0)	
Tumor Grade	*Grade 1*	7 (31.8)	5 (45.5)	2 (18.2)	0.170
	*Grade 2–3*	15 (68.2)	6 (54.5)	9 (81.8)	
Angiosarcoma Adjuvant Therapy					
Adjuvant Chemotherapy		10 (45.5)	5 (45.5)	5 (45.5)	--
Adjuvant Chemotherapy Regimen	*Doxorubicin/Ifosfamide*	4 (20.0)	2 (40.0)	2 (40.0)	--
	*Gemcitabine/Docetaxel*	4 (20.0)	2 (40.0)	2 (40.0)	
	*Both*	2 (10.0)	1 (20.0)	1 (20.0)	
Adjuvant Radiation		4 (18.2)	3 (27.3)	1 (9.1)	0.269
Recurrence/Survival in Patients Diagnosed >5 year ago		(*n* = 15)	(*n* = 8)	(*n* = 7)	
Local Recurrence		6 (40.0)	3 (37.5)	3 (42.9)	0.833
Distant Recurrence		3 (20.0)	3 (37.5)	0 (0.0)	0.070
Any Recurrence		9 (60.0)	6 (75.0)	3 (42.9)	0.205
Recurrence-Free Survival in Months (median, IQR)		53.7 (7.3–45.7)	54.7 (13.1–53.4)	52.5 (32.1–147.5)	0.315
Overall Survival in Months (median, IQR)		85.2 (32.9–106.8)	80.7 (36.0–105.8)	90.4 (32.1–147.5)	1.000

Bold denotes significance at *p* < 0.05.

**Table 2 cancers-13-03814-t002:** Subgroup analysis of patients who underwent neoadjuvant chemotherapy (NAC).

Variables		Breast Angiosarcoma Type			
		**Combined**	**Primary (PAS)**	**Radiation-Associated (RAAS)**	
		**(*n* = 10)**	**(*n* = 5)**	**(*n* = 5)**	
		***n* (%)**	***n* (%)**	***n* (%)**	***p***
Sociodemographics					
Age (median, IQR)		54 (43–72)	43 (29–50)	72 (57–76)	0.206
Race	*Asian*	1 (10.0)	0 (0.0)	1 (20.0)	0.292
	*Black*	0 (0.0)	0 (0.0)	0 (0.0)	
	*White*	9 (90.0)	5 (100.0)	4 (80.0)	
Ethnicity	*Hispanic*	4 (40.0)	3 (60.0)	1 (20.0)	0.197
	*Not Hispanic*	6 (60.0)	2 (40.0)	4 (80.0)	
Health Insurance	*Medicaid*	0 (0.0)	0 (0.0)	0 (0.0)	0.513
	*Medicare*	3 (30.0)	1 (20.0)	2 (40.0)	
	*Private*	6 (60.0)	3 (60.0)	3 (60.0)	
	*Uninsured*	1 (10.0)	1 (20.0)	0 (0.0)	
Neoadjuvant Therapy					
Neoadjuvant Chemotherapy Regimen	*Doxorubicin/Ifosfamide*	5 (50.0)	3 (60.0)	2 (40.0)	0.282
	*Gemcitabine/Docetaxel*	3 (30.0)	2 (40.0)	1 (20.0)	
	*Single-Agent Paclitaxel*	2 (20.0)	0 (0.0)	2 (40.0)	
Chemotherapy Cycles (median, IQR)		4 (3–5)	5 (3–6)	4 (4–4)	0.167
Neoadjuvant Radiation		2 (20.0)	2 (40.0)	0 (0.0)	0.114
Pathology					
Radiographic Tumor Size at Diagnosis in cm (median, IQR)		5.4 (2.2–7.2)	7.2 (6.6–7.5)	2.2 (1.5–2.5)	0.206
Pathologic Tumor Size at Surgery in cm (median, IQR)		4.3 (2.5–8.0)	6.6 (4.5–8.0)	3.0 (2.0–4.0)	0.206
Pathologic Response	No Response	2 (20.0)	1 (20.0)	1 (20.0)	0.766
	Partial Response	5 (50.0)	3 (60.0)	2 (40.0)	
	Complete Response	3 (30.0)	1 (20.0)	2 (40.0)	

## Data Availability

The data presented in this study are available on request from the corresponding author. The data are not publicly available due to privacy concerns.

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
