# Peer review of "Clinical Outcomes for Primary and Radiation-Associated Angiosarcoma of the Breast with Multimodal Treatment: Long-Term Survival Is Achievable"

_cancers, 2021, doi:10.3390/cancers13153814_

Round 1

Reviewer 1 Report

Although the sample size for this study is small, this paper is well written. There are several aspects that would need some additional explanation/editing prior to this paper's acceptance for publication:

  1. Under the Introduction, last paragraph, the authors wrote "Given our institution serves as a high-volume ..." Not clear what OUR institution means. Need to be more specific.
  2. Under the Introduction, last paragraph, the authors wrote "... after NAC given our institution's ..." You need to spell out NAC given that it is the first time you are unsing this acronym in the paper (abstract excluded).
  3. Under Data Source and Patient Selection, the authors wrote "PAS and RAAS patients diagnosed between ..." You need to be very specific and explain from the begining that only female patients were included in your study. Also, are you counting patients or diagnoses? Need to make sure you make this specification clear.
  4. Under Data Source and Patient Selection, the authors wrote "Patients who did not receive all or part of their first course of treatment at our institution were excluded". Why? A clear explnation is needed.
  5. Under "2.3 Statistical Analysis" the authors wrote "... Mann–Whitney U tests were used to compare continuous, non-parametric variables". Are the variables or the statistics non-parametric? Needs correction.
  6. Under "2.3 Statistical Analysis" the authors wrote "... to allow for a minimum of 5-year follow-up". Why? A clear, referenced, explanation is needed here.
  7. On line 106, the authors wrote "... based on cross-sectional imaging" This is confusing: is this study cross-sectional or retrospective?
  8. Based on your Table 1, you have only 1 Asian individual in your study. The rest of them are either White or Hispanic. Why did you keep this individual in your study? Needs a clear explanation.
  9. Based on your Table 1, you have only 1 individual who received total mastectomy as initial breast surgery. Since this is not a case-study, why did you keep this individual in your study? A clear explanation is needed.
  10. Based on your Table 1, you have only 1 individual who is a current smoker, the rest are all former smokers. Why are you still using "smoking status" in your analyses? And what does this "former smoker" status reflect in terms of breast diagnoses/treatment?
  11. Based on your Table 1, you have only 1 individual with Definitive Surgery (for PAS or RAAS) as chest wall resection. Why did you keep this individual in your analyses? A clear explanation is needed.
  12. On line 87 the authors wrote "... fisher exact tests were used to compare categorical variables" Why Fisher's exact test instead of the Chi-square Exact test?
  13. What population is your sample reflective of? To whom can your results be generalized to, if any? A clear explanation is needed.

Author Response

Reviewer 1: Although the sample size for this study is small, this paper is well written. There are several aspects that would need some additional explanation/editing prior to this paper's acceptance for publication:

  1. Under the Introduction, last paragraph, the authors wrote "Given our institution serves as a high-volume ..." Not clear what OUR institution means. Need to be more specific.
    Thank you for the comment. This has been clarified with the following text: "Our National Cancer Institute-designated Cancer Center (Sylvester Comprehensive Cancer Center), is a high-volume sarcoma referral center. As a result, the primary objective of this study is to analyze local recurrence (LR), distant recurrence (DR), and survival outcomes in the setting of contemporary multimodality therapy for PAS and RAAS."
  2. Under the Introduction, last paragraph, the authors wrote "... after NAC given our institution's ..." You need to spell out NAC given that it is the first time you are unsing this acronym in the paper (abstract excluded).
    Thank you for the comment. This has been addressed with the following text: “The secondary objective is to evaluate pathologic complete response (pCR) rates after neoadjuvant chemotherapy (NAC) given our institutional preference to treat PAS and RAAS with NAC."

  3. Under Data Source and Patient Selection, the authors wrote "PAS and RAAS patients diagnosed between ..." You need to be very specific and explain from the begining that only female patients were included in your study. Also, are you counting patients or diagnoses? Need to make sure you make this specification clear.
    Thank you for the comment. The following text has been edited and currently reads: “Female patients with diagnoses of PAS or RAAS made between 2010-2020 were identified through our institutional tumor registry using International Classification of Diseases 9th edition codes (171.4 and 171.9).”

  4. Under Data Source and Patient Selection, the authors wrote "Patients who did not receive all or part of their first course of treatment at our institution were excluded". Why? A clear explanation is needed.
    Thank you for the comment. We agree that this should be further explained. The following text has been added: “Patients who did not receive all or part of their first course of treatment at our institution were excluded since complete clinical, treatment, and outcome data was not available in our electronic medical records."

  5. Under "2.3 Statistical Analysis" the authors wrote "... Mann–Whitney U tests were used to compare continuous, non-parametric variables". Are the variables or the statistics non-parametric? Needs correction.
    Thank you for the comment. The following is the modified text: “Fisher exact tests were used to compare categorical variables, and Mann–Whitney U non-parametric tests were used to compare continuous variables. These tests were utilized to account for the relatively small sample size.”

  6. Under "2.3 Statistical Analysis" the authors wrote "... to allow for a minimum of 5-year follow-up". Why? A clear, referenced, explanation is needed here.
    Thank you for the comment. We agree this needs to be clarified. Please see the added text: “Five-year follow-up was selected to evaluate if patients undergoing multimodality treatment could achieve long-term survival for PAS and RAAS, a sign of clinical progress in a disease where historical survival has been less than 18 months.2,4-8"

  7. On line 106, the authors wrote "... based on cross-sectional imaging" This is confusing: is this study cross-sectional or retrospective?
    Thank you for this observation. We were referring to cross-sectional imaging (e.g. CT/MRI scans). This has been clarified with the following edited text: “There was no difference in clinical tumor size between PAS and RAAS based on computed tomography or magnetic resonance imaging (4.2 vs. 2.8cm, p=1.00), (Table 1).”

  8. Based on your Table 1, you have only 1 Asian individual in your study. The rest of them are either White or Hispanic. Why did you keep this individual in your study? Needs a clear explanation.
    Thank you for the comment. We aimed to include all patients in an effort to present a comprehensive experience of patients cared for at our institution. The following text has been added to the methods section: “While some variables only had a few patients per category (e.g. one Asian patient, one Medicaid patient, one patient with total mastectomy as initial operation, or one chest wall resection at definitive surgery), the authors chose to not exclude any patients in order to present the complete experience of patients treated at this center during the study period.”

  9. Based on your Table 1, you have only 1 individual who received total mastectomy as initial breast surgery. Since this is not a case-study, why did you keep this individual in your study? A clear explanation is needed.
    Thank you for the comment. A similar edit was utilized to answer your 8th inquiring with the following text: “While some variables only had a few patients per category (e.g. one Asian patient, one Medicaid patient, one patient with total mastectomy as initial operation, or one chest wall resection at definitive surgery), the authors chose to not exclude any patients in order to present the complete experience of patients treated at this center during the study period."
  10. Based on your Table 1, you have only 1 individual who is a current smoker, the rest are all former smokers. Why are you still using "smoking status" in your analyses? And what does this "former smoker" status reflect in terms of breast diagnoses/treatment?
    Thank you for the comment. Smoking status does not impact the diagnosis or treatment of this disease and has therefore been removed from the table.
  11. Based on your Table 1, you have only 1 individual with Definitive Surgery (for PAS or RAAS) as chest wall resection. Why did you keep this individual in your analyses? A clear explanation is needed.
    Thank you for the comment. A similar edit was utilized to answer your 8th inquiring with the following text: “While some variables only had a few patients per category (e.g. one Asian patient, one Medicaid patient, one patient with total mastectomy as initial operation, or one chest wall resection at definitive surgery), the authors chose to not exclude any patients in order to present the complete experience of patients treated at this center during the study period.”
  12. On line 87 the authors wrote "... fisher exact tests were used to compare categorical variables" Why Fisher's exact test instead of the Chi-square Exact test?
    Thank you for the comment. The following text is now present: “Fisher exact tests were used to compare categorical variables, and Mann–Whitney U non-parametric tests were used to compare continuous variables. These tests were utilized due to the relatively small sample size.” We cannot assume normality of the data so need to use non-parametric tests given the small sample size.
  13. What population is your sample reflective of? To whom can your results be generalized to, if any? A clear explanation is needed.
    Thank you for the question. We have added the following text to the discussion: "Additionally, this study is limited in its generalizability as it was conducted at an NCI-designated cancer center with a high-volume of sarcoma patients treated in a multidisciplinary setting. While the conclusions garnered from this study may be helpful to physicians at other centers, results may be difficult to replicate without a multidisciplinary sarcoma team."

Reviewer 2 Report

The manuscript entitled “Clinical Outcomes for Primary and Radiation Associated Angiosarcoma of the Breast with Multimodal Treatment: Long-term Survival is Achievable” by Kronenfeld et al shows better clinical outcome when patients with primary or radiation-associated angiosarcoma were provided with multimodality treatment regimen including adjuvant chemotherapy and radiation in addition to surgical resection. The authors report that most of the patients who received neoadjuvant chemotherapy had partial or pathologic complete response. This study is clinically significant and will contribute to the existing body of literature. However, there are some major and minor concerns that needs to be addressed.

Major Comments:

  1. Authors have not provided information on blood glucose and insulin levels on these patients. Hyperglycemia exacerbates various cancers including that of the breast. Thus, the authors must provide information on the diabetic profile.
  2. Authors have not provided information on whether they have reached menopause, and if there is any correlation between the menopausal status and disease outcome. Authors must address that.
  3. Authors must provide details on inflammatory profile of these patients, particularly in patients with radiation-associated breast angiosarcoma as they are older. Older patients have chronic low grade inflammation which could influence the disease outcome.

Minor Comments:

  1. Explain briefly what angiosarcoma of the breast is.
  2. Explain the following abbreviations: CCI, IQR and RFS.

Author Response

Reviewer 2: The manuscript entitled “Clinical Outcomes for Primary and Radiation Associated Angiosarcoma of the Breast with Multimodal Treatment: Long-term Survival is Achievable” by Kronenfeld et al shows better clinical outcome when patients with primary or radiation-associated angiosarcoma were provided with multimodality treatment regimen including adjuvant chemotherapy and radiation in addition to surgical resection. The authors report that most of the patients who received neoadjuvant chemotherapy had partial or pathologic complete response. This study is clinically significant and will contribute to the existing body of literature. However, there are some major and minor concerns that needs to be addressed.

Major Comments:

  1. Authors have not provided information on blood glucose and insulin levels on these patients. Hyperglycemia exacerbates various cancers including that of the breast. Thus, the authors must provide information on the diabetic profile.
    Thank you for the comment. We have now included information in Table 1 as to proportion of patients with diabetes. We have also included median blood glucose levels.

  2. Authors have not provided information on whether they have reached menopause, and if there is any correlation between the menopausal status and disease outcome. Authors must address that.
    Thank you for the comment. We have now included menopause information in Table 1. Likely due to our small sample size, there is no correlation between menopause and disease outcomes. The following text has been added to the discussion: “We are further limited in the small number of cases and likely underpowered statistical analyses, limiting our ability to effectively perform subgroup analyses (e.g. the impact of menopause on patient outcomes).”

  3. Authors must provide details on inflammatory profile of these patients, particularly in patients with radiation-associated breast angiosarcoma as they are older. Older patients have chronic low grade inflammation which could influence the disease outcome.
    Thank you for the comment. Unfortunately, inflammatory profiles were not collected on these patients, so we cannot comment on this information. We do, however, acknowledge the potential impact of inflammation on breast angiosarcoma, and this is important to include in laboratory profiles of future patients.

Minor Comments:

  1. Explain briefly what angiosarcoma of the breast is.
    Thank you for the comment. The following text has been added to the introduction: “Primary angiosarcoma of the breast (PAS) and radiation-associated angiosarcoma of the breast (RAAS) are rare sarcomas that affect the inner lining of blood vessels in the breast with an incidence of 0.05% and 0.02%, respectively [1-3].”

  2. Explain the following abbreviations: CCI, IQR and RFS.
    Thank you for the comment. We have added definitions to the text for CCI (a risk score predictive of mortality based on specified comorbidities), IQR (interquartile range), and RFS (recurrence free survival – time from completion of primary treatment to recurrence).

Round 2

Reviewer 1 Report

The authors addressed satisfactorily all my comments.